# Degradation-Dependent Stress Relaxing Semi-Interpenetrating Networks of Hydroxyethyl Cellulose in Gelatin-PEG Hydrogel with Good Mechanical Stability and Reversibility

**DOI:** 10.3390/gels7040277

**Published:** 2021-12-20

**Authors:** Kamol Dey, Silvia Agnelli, Elisa Borsani, Luciana Sartore

**Affiliations:** 1Department of Applied Chemistry and Chemical Engineering, Faculty of Science, University of Chittagong, Chittagong 4331, Bangladesh; 2Department of Mechanical and Industrial Engineering, University of Brescia, 25123 Brescia, Italy; silvia.agnelli@unibs.it; 3Department of Clinical and Experimental Sciences, University of Brescia, 25123 Brescia, Italy; elisa.borsani@unibs.it

**Keywords:** hydrogel, semi-IPN, gelatin, hydroxyethyl cellulose, PEG, cyclic compression test, stress relaxation, cartilage tissue engineering

## Abstract

The mechanical milieu of the extracellular matrix (ECM) plays a key role in modulating the cellular responses. The native ECM exhibits viscoelasticity with stress relaxation behavior. Here, we reported the preparation of degradation-mediated stress relaxing semi-interpenetrating (semi-IPN) polymeric networks of hydroxyethyl cellulose in the crosslinked gelatin-polyethylene glycol (PEG) architecture, leveraging a newly developed synthesis protocol which successively includes one-pot gelation under physiological conditions, freeze-drying and a post-curing process. Fourier transform infrared (FTIR) confirmed the formation of the semi-IPN blend mixture. A surface morphology analysis revealed an open pore porous structure with a compact skin on the surface. The hydrogel showed a high water-absorption ability (720.00 ± 32.0%) indicating the ability of retaining a hydrophilic nature even after covalent crosslinking with functionalized PEG. Detailed mechanical properties such as tensile, compressive, cyclic compression and stress relaxation tests were conducted at different intervals over 28 days of hydrolytic degradation. Overall, the collective mechanical properties of the hydrogel resembled the mechanics of cartilage tissue. The rate of stress relaxation gradually increased with an increasing swelling ratio. Hydrolytic degradation led to a marked increase in the percentage dissipation energy and stress relaxation response, indicating the degradation-dependent viscoelasticity of the hydrogel. Strikingly, the hydrogel maintained the structural stability even after degrading two-thirds of its initial mass after a month-long hydrolytic degradation. This study demonstrates that this semi-IPN G-PEG-HEC hydrogel possesses bright prospects as a potential scaffolding material in tissue engineering.

## 1. Introduction

Developing a simple synthesis protocol to design biomaterial capturing a compositional, structural, mechanical and biological resemblance to natural tissue is still a major challenge in the field of tissue engineering [1,2]. Despite significant progress in biomaterial designs, there are very few successes that truly demonstrate the biomaterials’ ability to copy the native biomechanical environment with high fidelity within one single synthetic platform that are truly cost-effective and clinically relevant [3,4,5]. Moreover, types of tissues and their structural and mechanical complexities make the biomaterial designs more challenging [2]. This is particularly true for regenerating articular cartilage using scaffold-based tissue engineering. The complex structural organization, unique load-bearing capacity and specific dynamic microenvironment (time-dependent viscoelastic nature) of cartilage tissue, those we are still unable to reproduce into a single scaffold with high fidelity, make cartilage tissue regeneration elusive [6,7]. The composition of cartilage—water (70–80%), collagen (50–75%) and glycosaminoglycans (15–30%)—provides it viscoelastic, compressive, tensile and frictional properties that are essential for its smooth function within the biomechanically arduous joint environment [6]. This necessitates an in-depth study on mechanical properties (long-term static plus dynamic properties) of the scaffolding materials intended to be used for in vitro cartilage tissue regeneration using stem cells.

Various types (natural, synthetic or hybrid) and forms (films, sponge, microsphere) of scaffolding materials can be used for the in vitro regeneration of cartilage tissue [7]. Among them, hydrogel—a highly hydrated polymeric crosslinked three-dimensional (3D) network—has attracted ample interest as a scaffolding material due to some of its exceptional properties such as high-water content, stability, flexibility, biocompatibility, degradability, similarity to natural tissue, tunability in physical, chemical and mechanical properties, as well as porosity required to impart a chondrosupportive and/or chondroinductive milieu [8]. Over the past few decades, semi-IPN networks of hydrogels were widely used for cartilage tissue engineering [9,10,11,12]. Its advantages include ease fabrication and marked mechanical and thermal properties, among others. As a constituent, numerous natural polymers such as collagen, gelatin, alginate, chitosan, cellulose, hyaluronic acid, etc. are being used for preparing semi-IPN hydrogel [13]. These natural polymers are generally biocompatible and non-toxic, which are the key requirements for tissue engineering [13]. However, they are less stable and possess poor mechanical properties compared to their counterpart, synthetic polymers [13]. In this scenario, a hybrid hydrogel consisting of both natural and synthetic polymers, exploiting synergy between biological compatibility and physiological relevance of natural polymers and desired mechanical properties, stability, reproducibility, tunability and processability of the synthetic polymers, can be an optimal choice of materials [7,14]. Additionally, a multicomponent hydrogel appears to mimic the composition of cartilage tissue closely [6].

Gelatin (G), obtained from different collagen sources by alkaline or acid processing [15], has been extensively used in the field of tissue engineering due to its various desirable features including biocompatibility, biodegradability, low cost and ease of manipulation [16]. Additionally, in contrast to collagen, gelatin shows low/no antigenicity, while its chemical composition is very similar to collagen, possessing cell-binding sites (e.g., arginine-glycine-aspartic acid (RGD) peptide) and enzyme-mediated degradation sites in its backbone [17]. More importantly, it is generally recognized as safe (GRAS) by the US Food and Drug Administration (FDA) for food processing, which could facilitate overcoming some stringent regulatory hurdles. Hydroxyethyl cellulose (HEC), a water-soluble cellulose derivative, is a non-ionic polysaccharide [18]. Chemically, its structural similarity to glycosaminoglycans [18], non-ionic nature, water solubility, biodegradability and β-glucose linkage make it a suitable candidate for tissue engineering applications [19,20]. On the other hand, polyethylene glycol (PEG) is the most studied synthetic polymer as a hydrogel in the field of tissue engineering because of its high hydrophilicity, excellent biocompatibility, limited immunogenicity, antigenicity, available active sites for chemical modification, intrinsic resistance to protein binding and cell adhesion [21].

Earlier, we reported the preparation (less porous) and cell viability of the G-PEG-HEC hydrogel [20] and the effects of gamma sterilization on the physicomechanical and thermal properties of the G-PEG-HEC hydrogel [22]. All these studies showed that the composition of the G-PEG-HEC hydrogel was cell-friendly and could be effectively and efficiently sterilized using 25 kGy gamma sterilization (a paramount requirement for cell culture and commercial use). As a further contribution, we report here on the preparation of highly porous and structurally stable semi-IPN G-PEG-HEC hydrogel using poly(ethylene glycol)diglycidyl ether (PEGDGE) as a crosslinking agent following gelation, freeze-drying and the post-curing process. In this present study, we particularly place emphasis on the variation in viscoelastic properties—key parameters include hysteresis loop area, percentage dissipation energy and stress relaxation behavior—of the hydrogel during one month-long hydrolytic degradation. The ability of the scaffold to dissipate cell-induced forces, referred to as stress-relaxation, is a key mechanical signal influencing stem cell fate and function [23,24]. Within tissue, cells always go through a dynamic microenvironment [24]. Numerous recent works revealed how time-dependent response (viscoelasticity) of scaffolds modulated stem cell behavior and stated that faster stress relaxing scaffolds promoted stem cell proliferation and differentiation towards specific tissue [5,23,24,25,26,27,28,29]. All these outcomes warrant an in-depth study of mechanical properties of the hydrogel before a biological test using stem cells and motivated performance of the present study. To the best of our knowledge, this is the first report of its kind presenting the variation in viscoelastic properties of the hydrogel during the month-long hydrolytic degradation.

## 2. Results and Discussions

### 2.1. Preparation of Hydrogel and Physical Properties

The semi-IPN G-PEG-HEC hydrogel was prepared using a simple and easy synthetic method under physiological conditions. The effective mixing of the components is important to produce a homogeneous structure without phase separation. The chemical or physical crosslinking is required for obtaining a three-dimensional structurally stable hydrogel. PEG diglycidyl ether may react with available = NH, -NH_2_, COOH and –OH functional groups. The addition of functionalized PEG into gelatin solution (a weight ratio of G to PEGDGE is about 4:1) involved the cross-linking/grafting process mainly between end epoxide groups of functionalized PEG and free lysine α-amino groups of the gelatin chain [30]. Both grafting and cross-linking may take place during the reaction of gelatin and functionalized PEG. However, grafting predominates the cross-linking during this reaction probably due to the low amount of lysine groups present in the gelatin chains (less than 5 wt% according to the analytical data provided by the supplier) which resulted in insufficient cross-linking and consequently quickly soluble products. Therefore, we used an excess amount of functionalized PEG over free amino groups’ content in gelatin followed by the addition of a specific basic curing agent EDA to impart a more cross-linking reaction which resulted in a more stable hydrogel network. The incorporation of HEC into the G-PEG crosslinked system introduced a semi-interpenetrating network (Figure 1), since no gelation was observed between only functionalized PEG and HEC in the same condition. We hypothesized that HEC combined and interpenetrated into the G-PEG crosslinked network through physical interactions (Figure 1). The recipe of the hydrogel (Table 1) was chosen to match the composition of cartilage tissue closely.

We performed FTIR analysis to investigate the hydrogel structure. The hydrogel was thoroughly washed to remove the un-reacted reagents and soluble components and freeze dried before FTIR analysis. Figure 1 shows the FTIR spectra of pristine functionalized PEG, pristine G, pristine HEC and G-PEG-HEC hydrogel. All characteristic peaks of pristine components were present in the final hydrogel, indicating a hybrid hydrogel. Pure gelatin shows the characteristic peaks at 1631 and 1552 cm^−1^, corresponding to amide I and II, respectively. The amide I peak is originated from C=O stretching vibrations and amide II from –NH bending vibrations and C-N stretching vibrations [20]. The spectrum of virgin HEC shows absorbance at 3390 cm^−1^ corresponding to the O-H stretching vibration. Peaks at 2921 and 2875 cm^−1^ represent a C-H aliphatic stretching vibration. The absorption band at 1456 cm^−1^ corresponds to a C–H bending vibration. The intense peak at 1054 cm^−1^ is due to C-O-C stretching vibration. More remarkably, in the case of the G-PEG-HEC hydrogel, more intense peaks observed around 2860–2930 cm^−1^ (ascribed to aliphatic C-H asymmetric and symmetric stretching vibrations) and 1066 cm^−1^ (attributed to C-O-C stretching vibrations) confirm the existence and entrapment of HEC polymeric chains into the G-PEG crosslinked/grafted network forming the final semi-IPN G-PEG-HEC structure [31].

The apparent density and porosity of the hydrogel were measured using the ethanol displacement method, and the results are shown in Table 1. The apparent density, porosity and gel fraction of the hydrogel were found to be 0.120 ± 0.01 (g/cc), 83.09 ± 4.4% and 82.54 ± 1.9%, respectively. Notably, hydrophilicity of hydrogels is one of the important features for tissue engineering application. The swelling nature of hydrogels can influence the pore size as well as diffusive and mechanical properties [32]. Accordingly, the equilibrium swelling ratio (%) of the hydrogel was evaluated by soaking the samples in distilled water at 37 °C for 72 h, and it was found to be 720.00 ± 32.0%. This higher swelling ratio is beneficial for tissue engineering application. The higher swelling ratio of the semi-IPN G-PEG-HEC hydrogel might be due to the presence of porosity (open interconnected pores) as well as the cellulose derivative, which is known as a superabsorbent polymer.

### 2.2. Morphology of Hydrogel

We noticed the formation of microporous compact thin skin after the lyophilization process, as shown in Figure 2a. Notably, this skin also easily peeled off using a razor. Figure 2a shows the staining microscopic image of the swollen sample showing skin (outer part endowed with closed pores) and the interior part (endowed with open and well interconnected pores) of the structure. The skin, composed of denser micropores, might be formed due to the interfacial tension between solid and air during lyophilization. However, the interior part (immediately below skin) showed the open and interconnected porous structure suitable for the cell culture as observed from staining (Figure 2a), optical (Figure 2b) and SEM (Figure 2c) micrographs.

### 2.3. Mechanical Properties of Semi-IPN Hydrogel

#### 2.3.1. Tensile Mechanical Properties

A tensile test was conducted using dumbbell-shaped samples in a wet condition equipped with a 50N load cell. The tensile modulus, strength and elongation at break were found to be 0.120 ± 0.001 MPa, 0.140 ± 0.03 MPa and 56.10 ± 8.6%, respectively, as tabulated in Table 2. The obtained tensile modulus (Young’s modulus) was equivalent to that of native articular cartilage (0.1–2.0 MPa). Importantly, the tensile stress-strain behavior of the hydrogel was non-linear (J-shaped), similar to those found for the natural extracellular matrix, as shown in the Figure 3.

#### 2.3.2. Cyclic Compressive Properties

Mechanical performance of the hydrogel is one of the critical criteria in designing scaffolds for tissue engineering applications. Different natural tissues have different mechanical properties: stiffness varies from the Pa (brain) to kPa (liver) to MPa (cartilage, bone) range [5]. As such, compression testing of the hydrogel was conducted in a swollen state (after immersion in distilled water for 48 h) to calculate the modulus and other important mechanical characteristics. The compressive modulus, strength and percentage dissipation energy measurements of the hydrogel were 0.06 ± 0.003 MPa, 0.09 ± 0.01 MPa and 15.67 ± 0.78%, respectively (Table 2). Cyclic compressive stress-strain curves were also non-linear, matching with stress-strain curves found for natural tissues. Cyclic compressive results showed tiny hysteresis loops expressing hydrogels’ elastic nature, as shown in Figure 4a. The hysteresis loop area represents the dissipation energy per unit volume upon deformation. From the second compression cycle on, all subsequent cycles followed the nearly similar hysteresis loops, indicating good reversibility and preservation of its elastic nature even undergoing 50% deformation of its original height. Moreover, the stress-time curves for the corresponding tenth cycles also illustrated the nearly constant stress values over time during the test (Figure 4b). Of note, the surprising result was that the hydrogel fully recovered to its original shape after undergoing such a high level of repeated compressions (Figure 5a). Taken together, a slight decrease in stress and no residuary strain during the consecutive loading-unloading cycles demonstrated elasticity, an excellent shape-recovery property and good mechanical stability of the hydrogel.

In case of cyclic compression, the hydrogel dissipated higher energy during the 1st cycle compared to other cycles (Figure 4c), which indicated that the 1st loading caused some damage into the hydrogel structure. The dissipation energy slightly decreased from the 2nd cycle to the subsequent 7th cycle. However, there was no significant variation in dissipation energy beyond the 7th cycle to 10th cycle, indicating achieving the mechanical stability. Moreover, there was no significant difference among percentage dissipation energies of the 5th and all subsequent cycles (Figure 4d). This implied that the dissipation energy (hysteresis loop area) and percentage dissipation energy (ratio of area bound between loading unloading curves to area under loading curve) calculated from the loading-unloading cycles also validated the reversibility and mechanical stability of the hydrogel after a few compression cycles.

Figure 6a,b shows the hysteresis loops of the hydrogel, dissipation energy and percentage dissipation energy with increasing the maximum strains from 20 to 50% strain. The hysteresis loops area became larger with increasing the maximum strain (Figure 6a). As evident from Figure 6b, the dissipation energy remained almost constant with increasing the maximum strains, whereas the percentage dissipation energy exponentially increased with the increasing strain. The increased hysteresis loops of the hydrogel at different maximum strains and the corresponding exponentially increased percentage dissipation energy indicate the increased resistance of the hydrogel to the deformation at higher strain levels.

#### 2.3.3. Monitoring Mass Loss and Mechanical Properties during Hydrolytic Degradation

Covalently crosslinked matrix degradation by hydrolysis or cell mediated cleavage is a key component along with stiffness in regulating cell morphology and differentiation in a 3D microenvironment [33]. Degradation alters the stiffness, swelling ratio and mesh size of the hydrogel. Consequently, we assessed the degradation phenomenon of the hydrogel by monitoring the gravimetric mass loss, incubating the sample at 37 °C in distilled water up to one month. In addition, we investigated the influence of degradation on the mechanical integrity of the hydrogel, and the results are illustrated in Table 3. The correlation among stiffness, strength and mass loss during degradation was established in Figure 7a. Mass loss was steadily increased during the incubation period judging the hydrolytically degradable hydrogel. After a month-long incubation, the hydrogel lost 65% of its original mass. During hydrolysis tenure, the crosslinked network breaks down into a smaller fragmentation of low molecular weight polymeric chains, subsequently, dissolving or washing out of the network, resulting in increased mass loss. This phenomenon reduced the crosslinking density and an increased amount of loose polymeric chains which ultimately lowered the mechanical stiffness and strength during degradation time (Figure 7a). The compressive stress-strain curves obtained at each time interval during the degradation period showed a successively decreasing trend of both stiffness and strength (Figure 7b). Figure 7c showed the relative stiffness/strength and remaining mass of the hydrogel as a function of degradation time. As evident, both stiffness and strength were decreased about 55% after 7 days of hydrolytic degradation. In addition to degradation, we suggest that this initially increased reduction of stiffness and strength could be a combined effect of two factors: (i) the as-prepared hydrogel initially lost more mass due to leasing out of eventually un-reacted reagents and dissolving components, resulting in an increased swelling ratio, and (ii) the swelling ratio attained a plateau region after 7 days of incubation (shown in Figure 8c later), indicating a saturated hydrogel by bound and free water, resulting in a pronounced softer gel compared to the initial condition. At this stage, the softness of the polymeric chains dominantly derived from bound water, which afterwards, remained nearly constant throughout the degradation period. As shown in Figure 7c, after one month-long degradation, the hydrogel retained about 18% and 15% of its original stiffness and strength, respectively, whereas 35% of its initial mass was remained, demonstrating a highly hydrolysable hydrogel. Importantly, regardless of being highly vulnerable to hydrolytic degradation, the hydrogel was structurally stable enough to handle for mechanical testing, indicating a faster surface degradation over the bulk degradation mechanism. During surface degradation, the degraded shorter polymeric chains (soluble oligomers) escaped from the surface of the hydrogel into the surrounding aqueous media, while the shorter polymeric chains generated from the bulk degradation process remained entrapped in the core of the hydrogel.

Figure 7d shows the successive loading/unloading curves obtained from the cyclic compression test at a set strain of 50% after one month-long degradation of the hydrogel. Clearly, it was observed that the induced stress showed a transient phenomenon: During the 1st cycle, the induced stress was 0.020 MPa, but, during subsequent cycles, it was gradually decreased and became stationary, reaching to a constant value of 0.015 MPa after multiple cycles. This behavior demonstrated as stress softening (Mullins’ effect), which is characterized by a lower resulting stress for the same applied strain, often observed in soft-tissue materials, where such behavior is known as ‘preconditioning’ [34]. It revealed an occurrence in the internal micro-fracture during the first loading. In addition, macroscopically enhanced network cleavages made the hydrogel more vulnerable to micro-fracture during compression. However, after a few cycles, nearly identical hysteresis loops were observed, indicating good reversibility, and achieving mechanical stability of the hydrogel after a few cycles. The corresponding stress-time curves also expressed exhibiting Mullins’ softening effect in early stages followed by stabilized stress after a few cycles (Figure 7e).

The hysteresis loop area and corresponding calculated dissipation energy are used to characterize the viscoelastic nature of the hydrogel: the higher the dissipation energy, the greater the viscoelasticity [35]. Consequently, the percentage dissipation energies of the hydrogel were calculated from the fifth cycle, after which there were no significant differences among subsequent loading/unloading curves, at each specific degradation time interval, and they showed increased percentage dissipation energy over time (Figure 7f). This increasing trend of percentage dissipation energy with the increasing of hydrolytic degradation time indicated the degradation-dependent viscoelasticity of the hydrogel. We hypothesized that the increasing value of percentage dissipation energy might be due to more migration of water and physical interactions between fragmented polymeric chains (entrapped)—both phenomena imparted by the degradation. Hydrolytic degradation cleaved the crosslinked polymeric network, making more functional groups available for physical interactions and more entanglements among the entrapped shorter polymeric chains in the core of the hydrogel, as well as leashing out smaller fragmented polymeric chains, leaving more void spaces for water molecules. All these factors facilitated the viscoelastic behavior of the hydrogel. Strikingly, despite such a substantial hydrolytic degradation, the hydrogel maintained its dimensional stability and good mechanical reversibility without succumbing to ultimate mechanical failure. That is the structural beauty of this developed semi-IPN G-PEG-HEC hydrogel.

#### 2.3.4. Stress Relaxation Behavior of Hydrogel

To evaluate the viscoelastic nature of the hydrogel, we explored the time-dependent relaxation response of the hydrogel. In the stress relaxation test, a constant strain of 15% was applied quickly and held constant, and subsequently, the stress was measured as a function of time. Firstly, we examined the strain-enhanced stress relaxation behavior of the hydrogel. Since strain stiffening is induced from increasing strain, we carried out the stress relaxation test of the hydrogel holding at 15%, 30% and 45% strains subjected to compression testing, as shown in Figure 8a,b. Peak stress increased with increasing strain due to the densification effect of the porous hydrogel (Figure 8a). Normalizing the stress values with the initial stress value for each strain and presenting them as a function of time (log scale) provided an increased rate of relaxation (Figure 8b). Clearly, faster stress relaxation behavior was observed with increasing the maximum strain—natural tissue also exhibits similar stress relaxing behavior with an increasing deformation level [36]. This phenomenon might be explained by the poro-viscoelasticity effect (volumetric deformation) arising from water migration within the hydrogel [37].

To understand how water content impacted the viscoelastic behavior of the hydrogel, we demonstrated stress relaxation responses of the hydrogel at four points in the swelling process (at 5, 24, 72 and 168 h), as shown in Figure 8c. The stress relaxation gradually increased as the swelling proceeded to equilibrium at 7 days (Figure 8c). During this swelling period, peak stress decreased from 0.042 MPa at 5 h to 0.021 MPa at 168 h, while the swelling ratio increased from 358 ± 4% at 5 h to 842 ± 8% at 168 h. The relatively faster stress relaxation over time with increasing water uptake might be due to the coupling effects of (i) migration of more water, (ii) softer polymeric chains enhancing matrix flow and (iii) lesser stiffness [37].

Moreover, to decouple the effect of stiffness (initial elastic modulus) and softness on the stress relaxation response, we approached to assess the stress relaxation test at a shorter swelling time period (1 to 72 h) where there were no significant variations of stiffness. Figure 8d illustrates the stress relaxing behavior of the hydrogel at 1, 5, 15, 24 and 72 h while maintaining nearly constant stiffness (shown in inset) at those points. The degree of swelling ratio significantly increased from 314 ± 5% at 1 h to 720 ± 18% at 72 h. Strikingly, the stress relaxing response revealed relatively faster relaxation with increasing the water uptake which might be mainly ascribed to the migration of water through the porous network.

Finally, we investigated the stress relaxation behavior of the hydrogel after 14, 21 and 28 days of degradation, and the results are shown in Figure 8e. As expected, the higher the degradation time (more mass loss), the faster the stress relaxation rate. Clearly, the hydrogel exhibited enhanced stress relaxation (21 ± 1% at 1 day to 80 ± 2% at 28 days) with increasing the degradation time, indicating the degradation-dependent viscoelastic nature of the hydrogel. The hydrogel exhibited substantial hydrolytic degradation over time because of a gradual breakdown of covalently crosslinked networks. We suggest that more network rupture means more fragmented polymeric chains, resulting in enhanced physical interactions among oligomers. The reversible physical interactions—breaking and subsequent reforming of these physical interactions—during the compression test facilitate the faster relaxation. Moreover, being more liable to micro-fractures and more water retention due to degradation can be ascribed to faster stress relaxation. Taken together, the mechanical behaviors of the semi-IPN G-PEG-HEC hydrogel closely match with those of native cartilage tissue, and how these mechanical cues affect the chondrogenic differentiation of human mesenchymal stromal cells is our next work plan.

## 3. Conclusions

In summary, we successfully developed a degradation-mediated stress relaxing semi-IPN G-PEG-HEC hydrogel system with highly interconnected porosity. The hydrogel system showed outstanding structural stability as well as good mechanical stability and reversibility. Under a repeated compressive cyclic test, the hydrogel showed preconditioning, non-linear elasticity and characteristic hysteresis. Hydrolytic degradation promoted viscoelasticity with increased stress relaxation and percentage dissipation energy—a most welcome phenomenon necessary for remodeling the scaffolding material during chondrogenic differentiation. Overall, the collective mechanical behaviors were equivalent to those of cartilage tissue. Finally, this hydrogel system possesses promising potentiality as a scaffolding platform for cartilage tissue regeneration.

## 4. Materials and Methods

### 4.1. Materials

Type A gelatin (pharmaceutical grade, 280 bloom, viscosity 4.30 mPs), produced from pig skin, was purchased from ITALGELATINE, Santa Vittoria d’Alba, Italy. Poly(ethylene glycol)diglycidyl ether (PEGDGE) (molecular weight 526 Da) was supplied by Sigma-Aldrich Co., Milan, Italy. Ethylene diamine (EDA) was provided by Fluka, Milano, Italy. Hydroxyethyl cellulose (molecular weight ~90,000) was supplied by Sigma-Aldrich Co., Milan, Italy. All materials were used without further purification.

### 4.2. Synthesis of Semi-IPN G-PEG-HEC Hybrid Hydrogel

The G-PEG-HEC hybrid hydrogel was prepared following four steps, namely, gelation, liquid nitrogen freezing, freeze-drying and a post-curing process. Briefly, gelatin (6 g) was dissolved in 65 mL distilled water at 45 °C under mild magnetic stirring. An amount of 1.4 g PEGDGE was added drop-wise in the homogeneous gelatin solution followed by the addition of 70 mg EDA. After 5 min, the HEC solution (7 wt%, 25 gm) was added into the reaction mixture. The final reaction mixture was gently stirred at 45 °C for 1 h to obtain a homogeneous solution and then poured into the glass plate. The solution was allowed to form gel for 1 h at room temperature. The resulting gel was carefully peeled off and cut into rectangular bars (5 cm × 1cm × 1cm) and placed into a Pyrex crystallizing dish. The freezing was conducted by resting the crystallizing dish on the surface of an 8-cm-deep pool of liquid nitrogen, enabling freeze-casting at −196 °C. Freezing was assessed visually. During the evaporation of liquid nitrogen, the gel was frozen from the bottom to the top. The gel was incubated for 30 min at the freezing temperature to ensure complete freezing. Subsequently, the frozen gel was transferred to the freeze-dryer (lyophilizer), operating under vacuum at −60 °C, for the sublimation of ice crystals, resulting in a porous stable dried hydrogel. Finally, the porous hydrogel was post-cured at 45 °C for 2 h in the oven under vacuum to complete the crosslinking/grafting reactions. The freeze-dried hydrogel was washed several times with distilled water at 37 °C to remove the unreacted reagents and dissolved components subsequently and finally was freeze-dried in the lyophilizer. The recipe is represented in Table 1.

### 4.3. Gel Fraction

The gel fraction (polymer fraction that remains after washing) was calculated by washing the samples several times in distilled water at 37 °C for 24 h and adopting the following equation:$$\mathrm{Gel}\;\mathrm{fraction}\;(\%)=\frac{W_{d}}{W_{o}}\times 100$$
where *W_d_* is the dry mass after washing, and *W_o_* is the initial dry mass of the sample.

### 4.4. Apparent Density and Porosity Measurement

The ethanol displacement method was used to measure the apparent density and porosity of the dried hydrogel. Ethanol was chosen as the wetting agent because it could easily infiltrate into the pores of the hydrogel and did not induce any shrinkage or swelling. In addition, the hydrogel did not dissolve into ethanol at room temperature. Prior to experiment, the entrapped air of the hydrogel was removed under vacuum. A known mass of dried hydrogel W was immersed into a graded cylinder containing a known volume (*V*_1_) of ethanol, and the total volume (*V*_2_) of ethanol and hydrogel was recorded. The volume difference (*V*_2_ − *V*_1_) resembled the volume of the hydrogel skeleton. The hydrogel was carefully removed from the ethanol, and the residual volume (*V*_3_) of the ethanol was measured. Finally, the total volume of the hydrogel was calculated as *V* = *V*_2_ − *V*_3_. Each experiment was conducted using five samples, and the average value was presented with the standard deviation (SD).

The apparent density (ρ) of the hydrogel was evaluated using the following equation:$$\mathrm{Apparent}\;\mathrm{density}\;(\rho )=\frac{W}{V_{2}-V_{3}}$$

The porosity (ε) of the hydrogel was measured adopting the following equation:$$\mathrm{Porosity}\;(\varepsilon )=\frac{V_{1}-V_{3}}{V_{2}-V_{3}}$$

### 4.5. Equilibrium Swelling Ratio (%) of the Hydrogel

The swelling ratio (%) of the hydrogel was evaluated by soaking the pre-massed samples into distilled water at 37 °C over a period of 240 h. At each defined time interval, the wet sample was taken out from distilled water, excess surface water was removed by gently pressing with tissue paper, and mass was measured precisely using an electronic analytical balance. The wet mass of the hydrogel was gradually increased with increasing time over a period of 72 h, and beyond that nearly a plateau region was observed. The equilibrium swelling ratio (%) was considered when there was no significant change in the wet mass of the hydrogel. The equilibrium swelling ratio was determined as:$$\mathrm{Equilibrium}\;\mathrm{swelling}\;\mathrm{ratio}\;(\%)=\frac{W_{t}-W_{o}}{W_{o}}\times 100$$
where *W_t_* is the wet mass of the hydrogel at 72 h, and *W_o_* is the dry mass of the hydrogel. The dry hydrogel was obtained by air drying the swollen hydrogel for 48 h followed by vacuum oven drying at 40 °C for 24 h until a constant mass was observed. A swelling experiment was carried out using five samples to minimize error and was reported as a mean value (±SD).

### 4.6. Structural Characterization

Fourier transform infrared (FTIR) spectra were obtained on dry hydrogels using a Thermo Scientific, Nicolet iS50 FTIR spectrophotometer equipped with a PIKE MIRacle attenuated total reflectance attachment and recorded over a range of 400 to 4000 cm^−1^ at a resolution of 4 cm^−1^.

### 4.7. Morphological Analysis

The macro and micro structures of the hydrogel were analyzed by using both an optical microscope and scanning electron microscope (SEM). For optical microscopic visualization, the morphologies of the surfaces and textures were observed using a stereomicroscope (LEICA DMS 300) with reflected light. For SEM analysis, the samples were covered by sputter nano gold particles, and porous structures were examined using the SEM Cambridge scanning electron microscope, with an operating voltage of 10 kV.

### 4.8. Tensile Test

The tensile mechanical properties of the hydrogel were measured by uniaxial tensile tests using a universal testing system (INSTRON series 3366) equipped with a 50 N load cell at a cross-head rate of 10 mm/min. Specimens were cut from the sample disk into a dumbbell shape using a die according to ISO37 type 2. The dumbbell-shaped specimens were frozen in liquid nitrogen, lyophilized under vacuum and post-cured at 45 °C for 2 h. The actual width (W) and thickness (T) of the specimens in the narrow portion of the dumbbell were measured using an optical travelling microscope. The samples, having a gauge length of 25 mm (L_o_), were clamped at a larger distance and stretched until failure. At least five specimens were tested in a wet condition swollen at 37 °C in distilled water. Engineering stress was calculated by dividing the recorded force by the initial cross-sectional area (W*T). Engineering strain was calculated as the ratio of cross-head displacement to L_o_. The initial elastic modulus (stiffness) was determined from the slope of the initial linear segment of stress-strain curves. To find a precise tensile strength, considering the significant variation in the thickness of the specimen, tensile strength was taken as the maximum force divided by the minimum cross-sectional area of the specimen. The strain value corresponding to the maximum stress was considered as a strain at break (elongation at break). The tensile mechanical properties were represented as the mean ± SD.

### 4.9. Compression and Cyclic Compression Tests

The compressive mechanical properties of the hydrogel were measured using a universal testing system (INSTRON series 3366) in an unconfined compression mood between two impermeable parallel plates (Figure 5a). Prior to testing, the as-prepared samples were soaked in distilled water at 37 °C for 48 h. The dimension of the samples was measured using an optical travelling microscope. Before commencing the compression test, a load of 0.01 N was applied to ensure complete contact between the sample surface and plate. For the compression test, rectangular-shaped samples were compressed at a strain rate of 10 mm/min with a 50 N load cell up to a maximum 50% strain of the original heights. The cyclic consecutive loading-unloading test was carried out without waiting time for 10 cycles, after which there was no significant change in the curve shapes. At least five specimens were tested in the wet condition. Engineering stress was calculated by dividing the recorded force by the initial cross-sectional area. Engineering strain under compression was defined as the change in height relative to the original height of the freestanding specimen. The initial elastic modulus (stiffness) was calculated from the first compression cycle and defined as the slope of the compressive stress-strain curve within the range of 5–10% strain. The compressive strength was defined as the maximum stress at 50% strain.

For the successive cyclic loading-unloading experiment at different maximum strains (%), the specimen was first compressed (loaded) to a maximum strain of 20% and then relaxed (unloaded). Without giving resting time, the specimen was again compressed to 30% maximum strain and relaxed again. These loading-unloading operations were repeatedly conducted on the same specimen with an increased maximum strain up to 50%. The dissipated energies were calculated from the area enclosed by the stress-strain curves at each maximum stain.

The energy absorption of the hydrogel was derived from the cyclic compressive stress-strain relations. The hysteresis loop area, bounded by the loading-unloading curves, indicates the dissipated energy or adsorbed energy due to the viscous nature of the hydrogel. The compression energy (kJ/m^3^), total energy applied to the hydrogel during compression, is defined as the area enclosed by the loading curve and horizontal axis, while relaxation energy (kJ/m^3^) is the area bounded between the unloading curve and horizontal axis. The dissipation energy (kJ/m^3^) loss during the hysteresis cycle was calculated from the area bounded within the hysteresis loop. The percentage of dissipation energy (%) was determined by dividing the dissipation energy (kJ/m^3^) by the compression energy (kJ/m^3^).

### 4.10. Hydrolytic Mass Loss Evaluation and Monitoring Mechanical Properties

The pre-massed (*M_i_*) samples were incubated in distilled water at 37 °C over a month-long period. At regular intervals of 7, 14, 21 and 28 days, the samples were removed, rinsed with fresh water, air dried followed by being vacuum dried at 45 °C for 4 h, and finally dried mass (*M_f_*) was measured. The mass loss (%) was calculated using following equation:$$\mathrm{Mass}\;\mathrm{loss}\;(\%)=\frac{M_{i}-M_{f}}{M_{i}}\times 100$$

Hydrolytic degradation can affect the mechanical properties of the hydrogel. Cyclic compression and stress relaxation tests were performed over time (7, 14, 21 and 28 days) to monitor the mechanical integrity as well as to observe the variation in dynamic mechanical properties of the hydrogel during hydrolytic degradation.

### 4.11. Evaluation of Stress Relaxation Response

The stress-relaxation properties of the hydrogel were assessed by an unconfined compression test. Stress relaxation tests were performed at different time points (day 0 (1, 5, 15 h) 1, 3, 7, 21 and 24) by incubating the samples in distilled water at 37 °C. For evaluating stress-relaxation behavior, the samples were pre-conditioned by carrying out the cyclic compression at 50% strain up to 5 cycles until no significant changes in the curves as well as maximum stresses were observed. The pre-conditioned samples were quickly compressed to 15% strain with a deformation rate of 60 mm/min, and the strain was held constant, while the variation of stress was recorded as a function of time up to 20 min, as shown in Figure 5b,c.

## Data Availability

Data are contained within the article.

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
