# Peer review of "Degradation-Dependent Stress Relaxing Semi-Interpenetrating Networks of Hydroxyethyl Cellulose in Gelatin-PEG Hydrogel with Good Mechanical Stability and Reversibility"

_gels, 2021, doi:10.3390/gels7040277_

Round 1

Reviewer 1 Report

This manuscript reported a degradation-mediated stress relaxing semi-IPN G-PEG-HEC hydrogel. The hydrogel exhibited good structural and mechanical stability, as well as stress relaxation induced by hydrolytic degradation. Overall, the results and explanations in this manuscript is impressive and reasonable. There are some inquiries as follows:

  1. The Scheme 1 misleads readers. If freeze-drying is done, the hydrogel will be converted into aerogels or solids. However, the picture seems like a hydrogel before freeze-drying.
  2. Line 161, swelling ratio of other hydrogels for tissue engineering is suggested to be added, confirming that your swelling ratio is higher. Besides, is there a desirable range of swelling ratio for hydrogels in tissue engineering field? Because higher swelling ratio generally leads to poor mechanical properties after swollen.
  3. Authors said that the hydrogel showed great prospects in cartilage tissue engineering. However, there is no corresponding animal experiments to confirm it. If authors can not provide more data to support, the words “Cartilage Tissue Engineering” is suggested to be revised into “Potential Cartilage Tissue Engineering”.

Reviewer 2 Report

The paper entitled “Degradation-Dependent Stress Relaxing Semi-Interpenetrating Networks of Hydroxyethyl Cellulose in Gelatin-PEG Hydrogel for Cartilage Tissue Engineering” is very well written. The topic is substantial and interesting, the results show great potential.

Reviewer 3 Report

SUMMARY:

The authors reported a semi-interpenetrating (semi-IPN) polymeric hydrogel system as a degradation-mediated stress relaxing platform and performed comprehensive mechanical tests of the hydrogel system. The hydrogel showed extraordinary mechanical stability upto 1 month of in vitro incubation, which could be potentially applicable in cartilage tissue regeneration. Overall, the manuscript is well-written covering analysis and discussion on the results of mechanical tests. However, the title and overstatement in the manuscript could mislead the authors, so the reviewer recommend checking some minor points to improve quality of the manuscript.

Minor comments:

- The mechanical properties of the hydrogel system play crucial roles in successful cartilage regeneration as the authors stated. However, without any cellular or in vivo evaluation, it is not supportive to state the potential use of the semi-IPN G-PEG-HEC hydrogel in cartilage regeneration. The authors should remove cartilage tissue engineering from the title and modify it with focus on the content of this manuscript.

- Compressive property and shape-recovery of semi-IPN hydrogel could be interesting characteristics of this system. I would recommend replacing Figure 1 after Figure 3 with explanation in the main content.

- I would recommend to shorten the introduction section by removing unnecessary background information.
